# Perceived feasibility, facilitators and barriers to incorporating point-of-care testing for SARS-CoV-2 into emergency medical services by ambulance service staff: a survey-based approach

Kile Green ,[1] Massimo Micocci,[2] Timothy Hicks ,[1,3] Amanda Winter,[1,3] Joanne E Martin,[4] Bethany Shinkins,[5] Lisa Shaw,[6] Christopher Price,[7] Kerrie Davies,[8] Joy A Allen[1,3]

For numbered affiliations see end of article.

**Correspondence to**
Dr Kile Green;
Kile.Green@newcastle.ac.uk

## ABSTRACT

**Objectives** This body of work aimed to elicit ambulance service staff's perceptions on the barriers and facilitators to adoption, and clinical utility of incorporating rapid SARS-CoV-2 testing during ambulance assessments.

**Design** A mixed-methods survey-based project using a framework analysis method to organise qualitative data.

**Setting** Emergency and non-emergency care ambulatory services in the UK were approached to take part.

**Participants** Current, practising members of the UK ambulance service (paramedics, technicians, assistants and other staff) were included in this body of work.

**Results** Survey 1: 226 responses were collected between 3 December 2020 and 11 January 2021, 179 (79.2%) of which were completed in full. While the majority of respondents indicated that an ambulance-based testing strategy was feasible in concept (143/190, 75.3%), major barriers to adoption were noted. Many open-ended responses cited concerns regarding misuse of the service by the general public and other healthcare services, timing and conveyance issues, and increased workloads, alongside training and safety concerns. Survey 2: 26 responses were received between 8 February 2021 and 22 February 2021 to this follow-up survey. Survey 2 revealed conveyance decision-making, and risk stratification to be the most frequently prioritised use cases among ambulance service staff. Optimal test characteristics for clinical adoption according to respondents were; accuracy (above 90% sensitivity and specificity), rapidity (<30 min time to results) and ease of sample acquisition.

**Conclusions** The majority of commercially available lateral flow devices are unlikely to be supported by paramedics as their duty of care requires both rapid and accurate results that can inform clinical decision making in an emergency situation. Further investigation is needed to define acceptable test characteristics and criteria required for ambulance service staff to be confident and supportive of deployment of a SARS-CoV-2 test in an emergency care setting.

## STRENGTHS AND LIMITATIONS OF THIS STUDY

⇒ This body of work represents a mixed-methods survey-based approach to determine feasibility of SARS-CoV-2 testing in ambulances, capturing qualitative and quantitative outputs from front-line healthcare workers from all regions of England and Northern Ireland.

⇒ The inclusion of open-ended questions allowed for in-depth responses to be captured from respondents, providing additional context and reasoning.

⇒ Limited piloting of the surveys was performed due to the short timespan available to develop, disseminate, collect and analyse results, while the short interval of data capture (3 December 2020–11 January 2021) limited the reach and coverage of the survey.

⇒ The low (52%) response rate to the follow-up survey (survey 2) may have introduced a degree of bias and represents a much more limited dataset than survey 1.

## BACKGROUND

In response to the COVID-19 pandemic, there was large scale production of rapid, portable clinical tests for SARS-CoV-2 with CE (Conformitè Europëenne)-marking for in vitro diagnostics approval.[1 2] Like any test, their true value hinges on their role within a clinical pathway, the real-world accuracy of the test within that role and the efficacy of the decisions made following test results.[3 4] There has been significant debate about whether some of these tests are 'fit for purpose', for example, in the UK, the widespread use of rapid antigen-based lateral flow devices (LFDs) has raised concerns because their sensitivity has been shown to drop dramatically (from 73% to 48.8%) when carried out by non-trained healthcare workers.[5–7]

In an attempt to provide greater clarity on the minimum acceptable and desirable characteristics for rapid, point-of-care (PoC) tests for SARS-CoV-2, the WHO and the UK Medicines and Healthcare Products Regulatory Agency developed and published target product profiles (TPPs) for novel tests in this space.[8] However, given the vast number of use cases for PoC tests for SARS-CoV-2 and the ongoing need to provide guidance throughout the UK pandemic, the TPPs were reasonably generic. Further work is required to understand the exact role of these tests within specific settings and the potential downstream consequences of testing.[9]

Ambulance services have previously been highlighted as potential settings where rapid PoC testing, including for SARS-CoV-2, could add value for patients and the healthcare system, as they are frequently the first point of contact with healthcare services during an emergency.[10–12] However, due to the unique environment, including reliance on a dispersed workforce, operating under time pressures and with limited equipment,[13] it is necessary to identify setting-specific clinical scenarios where testing could effectively integrate into ambulance care processes. At the time of survey dissemination, at the height of the COVID-19 lockdown in the UK, some ambulance service trusts had begun to trial the use of PoC SARS-CoV-2 testing, but this was not widely performed. The objective of this body of work was to generate evidence and support decision making on the integration of SARS-CoV-2 PoC tests using a mixed-methodology approach to elicit views from ambulance service staff about factors that would facilitate or act as a barrier to PoC testing for SARS-CoV-2 in this setting. Collecting feedback and insights from end users of PoC devices would enable decision-makers to determine the feasibility and expected use cases in an ambulance setting, and work to mitigate any potential barriers to adoption to ensure the greatest added value to healthcare is achieved.

## METHODS
### Study design
This study consisted of two surveys, each following a mixed-methodology approach, designed to capture perceptions on feasibility and prioritisation of potential clinical scenarios where PoC SARS-CoV-2 tests could add value to healthcare. The surveys were developed by test evaluation methodologists (KG, TH, AW, MM and AJA), who are all experienced in qualitative methods for diagnostic evaluation, and were constructed using the online survey tool SurveyMonkey.[14] No questions were mandatory on either survey. No sensitive or patient identifiable data were included.

Respondents to survey 1 were invited to complete survey 2.

▶ Survey 1 (online supplemental material 1) was launched on 3 December 2020 and was open to responses until 11 January 2021. The reach of the survey is unknown due to the multiple methods of dissemination, however, 212 ambulance service respondents started the survey. Survey 1 aimed to identify key facilitators and barriers to the adoption of PoC SARS-CoV-2 tests in ambulances, with a focus on prioritising the key clinical scenarios where the tests could add value alongside the practical aspects that could affect adoption (environmental factors, test turnaround and usability).

▶ Survey 2 (follow-up) (online supplemental material 2) was disseminated to interested parties on 8th February 2021 and was open to responses until 22 February 2021. Fifty respondents to survey 1 were sent survey 2 via email. Out of 50, 26 (52%) respondents completed survey 2. This survey aimed to identify the most acceptable diagnostic accuracy requirements of rapid tests specifically to meet the prioritised clinical scenarios that were identified in survey 1.

### Setting
The surveys included in this body of work were conducted at the height of the UK pandemic lockdown period when daily cases were at the highest levels seen in the UK since wide-scale reporting began.[15] Some National Health Service hospital Trusts reported 100% capacity of critical care beds at this time and the matter was discussed extensively in media and during parliamentary sessions.[16] It is important to highlight this context in mind as it reflects the urgency at which the work was performed and affected the capacity for participants to respond to the survey, the content of the survey responses and the concerns raised at this time.

### Dissemination and participant selection
Participants were selected by purposive sampling with additional snowballing, targeting ambulance service staff at all levels to gain breadth of viewpoints across the service. The primary questionnaire was disseminated via email by Ambulance Trust Research and Development (R&D) teams as well as through promotion of the survey via email (JEM) to professionals involved in rapid testing in the acute sector. Survey 2 was disseminated via email directly to those respondents to survey 1 who had stated that they would be interested in providing additional input through follow-up questionnaire. Participation was voluntary and anonymised.

### Purpose and structure
Survey 1 (online supplemental material 1) was structured with a mixture of multiple choice, Likert and free-text answers to capture the following:
1. Demographics, exposure/familiarity with SARS-CoV-2 tests.
2. Desired test characteristics in general (including accuracy, ease of use, time to result (TTR)).
3. Potential clinical scenarios where PoC testing for SARS-CoV-2 testing could add value.
4. Feasibility of integrating PoC tests into current processes and patient pathways.

Survey 1 included two open-ended questions: 'What do you think the main concerns of adding a test on board the ambulance would be?' and 'Is there anything else you feel is important to tell us about testing in ambulances?'.

In survey 2 (online supplemental material 2), respondents were asked to state whether three hypothetical tests with sensitivities and specificities of >95%, >80% and <80%, respectively, would be useful for each of the priority clinical scenarios identified in survey 1 (a Likert scale from 'extremely useful' to 'not useful at all').

To ensure respondents understood complex technical performance terminology such as sensitivity and specificity, multiple choice answers included an example description of the value, for example, 60% sensitivity (6/10 people with COVID-19 get a positive test result).

### Reporting guidelines

The Standards for Reporting Qualitative Research checklist was used to structure the methodology and results of this study alongside the 'CHERRIES' checklist for reporting of online surveys.[17 18]

### Data analysis

**Quantitative analysis of structured responses:** Data processing and generation of summary statistics was performed using Microsoft Excel. As all questions were optional, denominators for each question reflect the number of responses to that individual question. Graphics and visuals were produced using the 'CANVA' web-tool suite to display the aggregated results. For questions using a Likert scale, a mean summary score (with SD) was calculated based on the following mapping— most important=2, very important=1, important=0, less important = −1, not important = −2. Final scores were then ranked from highest to lowest.

**Thematic analysis of free text responses:** A framework analysis method[19] was used to organise data from the qualitative, open-ended questions. Thematic analysis aimed to identify common issues and concerns about testing within the ambulance service. Initial coding of the responses were performed independently by two researchers (KG and MM), with a third acting as adjudicator (AJA) to reduce bias and improve consistency. Codes were discussed internally to reach an agreement on definitions, content and to improve validity. Codes were then clustered under four main categories to identify cross-cutting patterns. Final themes and definitions were agreed collaboratively among the qualitative researchers.

### Patient and public involvement

The National Institute for Health Research Newcastle In Vitro Diagnostics Co-operative (MIC) Insight patient and public involvement and engagement (PPIE) panel were consulted at various stages of the project, from conception to closing. Comments were received on acceptability of questions, length of the survey and expected impact of participating in this work. Initial comments were gathered from the PPIE panel on questions exploring COVID-19 testing and interpretation of results. Their feedback helped to formulate and refine questions to better facilitate participant understanding. The PPIE panel suggested that additional information on the terms 'sensitivity' and 'specificity' be included in the survey. In addition, the PPIE leads wrote a lay summary of the project outputs for dissemination via the MIC website and social media.

## RESULTS

### Demographics

Of the 212 ambulance service respondents, 150 were listed as 'Paramedics' with 55 of those considered 'Experienced paramedics'—with more than 2 years of experience in the role. Other roles included technical, support and call-handling staff.

One hundred and fourteen (55.8%) of the ambulance staff respondents (hereafter referred to as 'respondents') were based in the East of England, with a further 46 (21.7%) based in the Greater London region (table 1). All regions of England and Northern Ireland were represented by at least one respondent.

### Feasibility of testing

The majority of respondents answering questions on feasibility stated that it would be feasible to test patients for SARS-CoV-2 before transfer to hospital with 143/190 (75.3%) stating 'high' or 'very high' feasibility, including 22/26 (84.6%) of the respondents who had previously noted prior experience of using LFDs for SARS-CoV-2 or other indications.

Twenty six out of 101 (24.7%) stated they had experience conducting tests for any respiratory infection in a prehospital setting, including (LFDs) and oxygen saturation monitors. In addition, blood glucose monitoring tests were already in use and routinely performed on route to secondary care.

### Use cases and sampling method

The top four ranked clinical scenarios where a PoC test for SARS-CoV-2 was deemed to add value, accounted for 92.4% of the total responses to this question (figure 1):
1. 'Triaging of patients prior to arrival at secondary care facilities to improve handover' (83/170, 48.8%).
2. 'Aid decision making on where a patient should be referred to next' (40/170, 23.5%).
3. 'Risk stratification of patients to determine if a patient can be safety left at home' (28/170, 16.5%).
4. 'Rationalising personal protective equipment (PPE) use for ambulance service staff' (6/170, 3.5%).

Other responses were free-text answers including 'all options', 'none of the options' and other niche uses.

Irrespective of other technology considerations, finger prick blood sampling was stated as the optimal sampling technique by one-third of the respondents (125/189,

**Table 1**  Summary table of ambulance service responders by region and job role

| Region | Ambulance service staff (total) | Paramedics and experienced paramedics | Technicians and assistants | Other staff |
|---|---|---|---|---|
| East Midlands | **5** | 3 | 2 | 0 |
| East of England | **114** | 72 | 35 | 7 |
| Greater London | **46** | 36 | 9 | 1 |
| North East | **2** | 2 | 0 | 0 |
| North West | **3** | 3 | 0 | 0 |
| Northern Ireland | **1** | 1 | 0 | 0 |
| South East | **13** | 10 | 3 | 0 |
| South West | **8** | 5 | 3 | 0 |
| West Midlands | **5** | 5 | 0 | 0 |
| Yorkshire and the Humber | **15** | 13 | 2 | 0 |
| Total | **212** | **150** | **54** | **8** |

Bold values represent the total row and column values.

66.1%). Twenty-nine (29/189, 15.3%) and 19 (19/189, 10.1%) respondents preferred nasopharyngeal swabbing and saliva sampling, respectively. Respondents using the 'other' option were largely in favour of testing, regardless of the specific method used.

**Importance of test characteristics:** 'Accuracy' and 'TTR' of the test were the most important characteristics to the responders with a mean adjusted score of 1.47 (SD=0.44) and 1.32 (SD=0.35), respectively (figure 2). 'Storage of the kit' (mean score 0.14) and 'size of the kit' (mean score −0.13) were among the least important characteristics. Additional comments provided by paramedics included 'minimisation of staff exposure time', 'room temperature storage' and 'cost'.

TTR 88/170 respondents (51.8%) stated a maximum acceptable TTR of 15 min or less, with 162/170 (95.3%) selecting 30 min or less.

**Sensitivity and specificity:** The majority of responses required over 80% sensitivity (56/170, 32.9% selected 'above 80% sensitivity', 73/170, 42.9% of responders selected 'above 90% sensitivity'). Similar responses were obtained for minimum specificity requirements (43/170, 24.7% respondents selected 'above 80% specificity', and 79/170, 46.5% selected 'above 90% specificity').

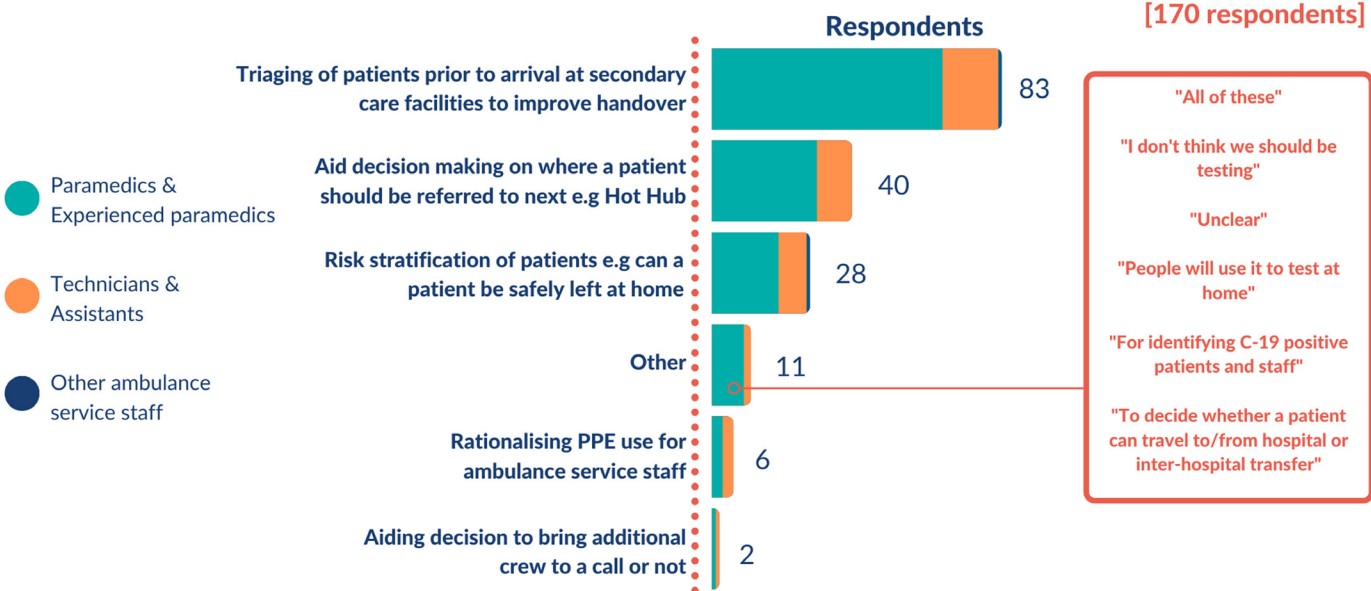

**Figure 1**  Summary of use cases for rapid SARS-COV-2 testing in ambulances. A total of 170 respondents stated their preferred use cases for a rapid SARS-CoV-2 test in the ambulance service as displayed in this stacked chart. The length of the bar represents the number of respondents providing that use case, with the colours representing the categories of ambulance service staff. A call-out text box displays the main quotes from respondents that noted 'other' as their preferred use case.

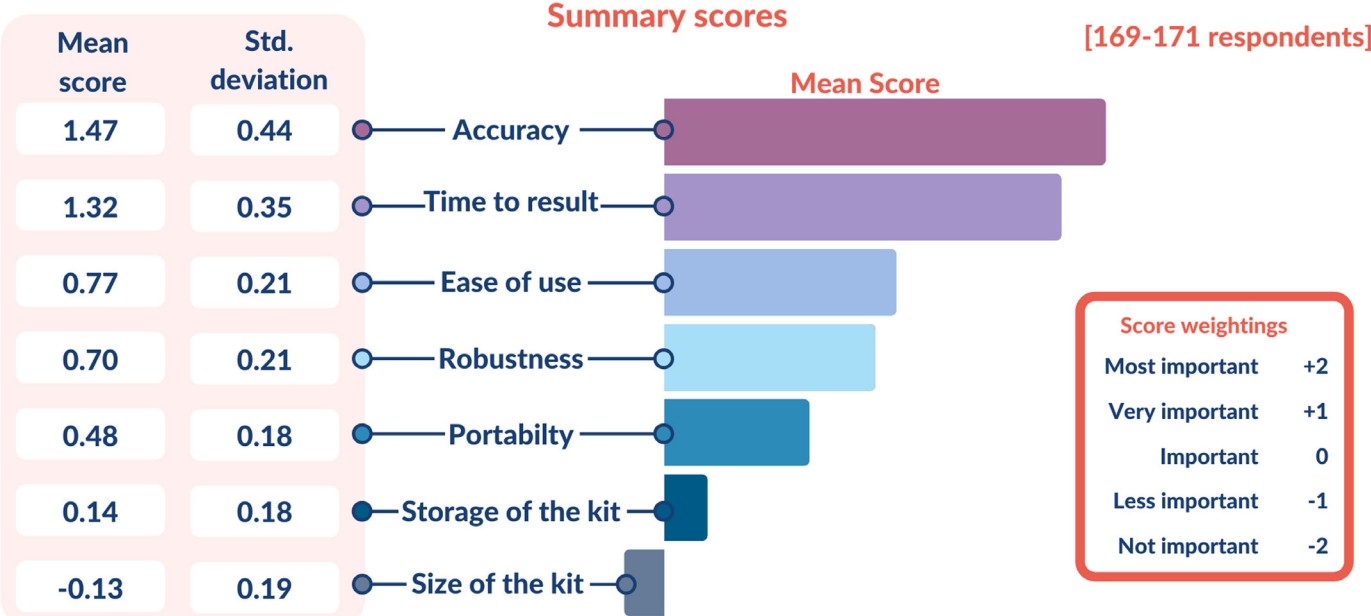

**Figure 2** Summary scores for most important test characteristics for SARS-CoV-2 testing in an ambulance setting. Summary scores for the Likert ranking of test characteristics for a rapid SARS-CoV-2 test in the ambulance service are displayed in this chart. Likert scores ranged from –2 (Not important) to +2 (Most important). mean score and SD of the results are displayed in the figure.

**Thematic analysis of qualitative responses:** A total of 11 subthemes were generated from the 128 responses to the two free-text questions from the primary survey (survey 1) and categorised into 4 broad themes (service-level implications, Direct impact on patient care, Practical use of the test and Safety) to identify cross-cutting patterns. Subtheme definitions have been includes in table format (table 2). Representative quotes in the results have been assigned a numerical identification code (IC) based on participant number.

### Category 1: service-level implications
#### Misuse of service
The major concern from respondents for the roll out of an ambulance based rapid SARS-CoV-2 test was the misuse of the service by the public, appearing in almost half of the responses —*'The general public will eventually find out and will call 999 just to get a test.'* [IC7] This concern also expanded into other service providers too—*'People will call 999 just to be tested. Also 111 & GPs will send us just to test. We should only be responding if symptoms suggest it is a 999 emergency'* [IC74]. Many felt that offering SARS-CoV-2 testing would result in increased call volume and a resulting increase in call-outs to patients in a non-critical situation *'distracting from the primary cause for 999 calls'* [IC158]. These concerns were commonly cited with fear of increased workload for ambulance service staff.

#### Workload
The potential for additional work to be added in the form of processing and performing tests for SARS-CoV-2 prompted some strong, negative responses commonly paired with concerns involving the misuse of the

ambulance service. Fundamentally, some service staff considered the SARS-COV-2 test *'something else to do'* [IC70] in an otherwise busy and time critical role. Concerns were raised about *'additional call outs from patients and other healthcare organisations as a means of rapid testing'* [IC81] leading to an impact on the efficiency *and 'the wider ambulance service resource demand'* [IC55].

#### Public expectation
Responses reflected on previous experience, with cases of the public not declaring disease or symptoms to avoid incorrect perceived stigma or reduced care—*'Patients have been omitting to tell crews about COVIDCOVID-19 symptoms as they feel they won't get an ambulance if they tell the truth'* [IC42]. Further concerns stemmed from the overarching belief that the general public may use the ambulance service as a more convenient way to get a SARS-COV-2 test—*'Do not advertise to the public'* [IC52], *'If testing were to be carried out by ambulances, attempts should be made to keep this away from public knowledge'* [IC97]. Other respondents suggested that the general public already believes that the ambulance services provides routine SARS-COV-2 testing *'people think we do it anyway'* [IC178], a concern that has already been noted as an issue for some ambulance trusts in previous waves.[20]

### Category 2: impact on patient care
#### Consequences to care
Confidence in rapid SARS-COV-2 tests appeared to be low with concerns over increased harm for patients and staff associated with erroneous results *'False positives/ negatives giving false sense of security/resulting in incorrect decisions being made based on that information'* [IC91].

**Table 2** Summary table of themes for thematic analysis of open-ended responses

| Theme | Sub-theme | Description | Examples |
|---|---|---|---|
| Service level implications | Misuse of service | Implications for 999 service and unnecessary paramedics call out (Public and other healthcare providers misusing the Ambulance service) | 'The general public will eventually find out and will call 999 just to get a test.'<br>'People will call 999 just to be tested. Also 111 & GPs will send us just to test. We should only be responding if symptoms suggest it is a 999 emergency'<br>'distracting from the primary cause for 999 calls' |
| Service level implications | Workload | Concerns regarding additional workload due to SARS-COV-2 testing | 'something else to do'<br>'additional call outs from patients and other healthcare organisations as a means of rapid testing'<br>'the wider ambulance service resource demand' |
| Service level implications | Manage public perception | Comments on managing public opinion or perception of the use of COVID testing in ambulances | 'Patients have been omitting to tell crews about COVID symptoms as they feel they won't get an ambulance if they tell the truth'<br>'people think we do it anyway' |
| Impact on patient care | Consequences to care | Impact on the system and on patient care (eg, use of PPE) | 'False positives/negatives giving false sense of security/resulting in incorrect decisions being made based on that information'<br>'Culture of wearing PPE would relax' |
| Impact on patient care | Use case to inform clinical decisions | Concerns around how/who/when/why to test, and effect or impact of the result on decision making or conveyance | 'Does the result really change the outcome especially if leaving someone at home?'<br>'Why do we need to test? We treat for what we see. If someone is very ill they go to hospital, if they aren't they could stay at home and be referred on. A COVID diagnosis doesn't make a difference' |
| Impact on patient care | Acceptance | Test acceptance and 'trust' on results | 'Patients have been omitting to tell crews about COVID symptoms as they feel they won't get an ambulance if they tell the truth' |
| Impact on patient care | Time implications | Delay to service or conveyance due to testing or time to results | 'ensuring it does not delay transfer, especially in time critical patients'<br>'delay will increase the time the patient will be with the driver and thus may affect the Key performance Indicators'<br>'increased job cycle times'<br>'delaying treatment or transport due to waiting for results'<br>'delays on scene'<br>'Some people wear level 3 PPE for any COVID+ve patient, meaning if they tested+ve they would then delay things further by going to change PPE' |
| Practical use of the test | Test feasibility, correct use and integration with ambulance setting | Usability issues, additional skillset required, appropriate staff training for use of the test and integration with ambulance setting. Storage, portability and stocking of the tests in the ambulance and a paramedic kit | 'How it will be carried - it needs to be introduced into standard response bags that are taken to every patient, similar to our blood sugar testing kits. No one will want to go back to the ambulance to collect a large, bulky box to do it'<br>'kits won't be stored correctly due to fluctuating temperatures of a vehicle outside in all weathers'<br>'keeping in a sterile place'<br>'Quantity & Supply in order to test volume of patients in a shift (~5–9 per 12 hour shift). Availability of tests for relatives'<br>'a naso swab could be difficult in a moving vehicle, test would need to be carried out prior to transport' |
| Practical use of the test | Integration into pathways | Test accuracy and time to results | 'rely on the test having very high specificity and less important but high sensitivity'<br>'That it is not accurate enough but hospitals will begin to rely on it and not follow up test'<br>'If it is not specific enough it could lead to clinicians making inaccurate decisions about patient care' |
| Safety | Risk exposure for staff members | Concerns and comments on risk of exposure to staff | 'Increased risk to staff carrying out [the] test'<br>'chance of using AGP's then we could use a L3 mask instead'<br>'That we will increase our on scene time with a positive patient' |
| Safety | Staff reassurance | Staff reassurances | 'would assist in reassurance for crews' |

GP, general practitioners; PPE, personal protective equipment.

Additional concerns were raised regarding staff adherence to PPE requirements—'*Culture of wearing PPE would relax*' [IC108].

## Informing clinical decisions

Overall, the added value of testing was questioned. Comments on who, when, where or why to test were captured, including—'*Does the result really change the outcome especially if leaving someone at home?*' [IC28], '*Why do we need to test? We treat for what we see. If someone is very ill they go to hospital, if they aren't they could stay at home and be referred on. A COVID diagnosis doesn't make a difference*' [IC94]. Multiple responses inferred that testing would not change decision making, referring to the needs of the patient above or beyond their SARS-CoV-2 status—'*Value in most patients limited, other clinical indicators rule COVID in or out, and if they are unable to cope at home they have to be conveyed*' [IC145].

## Acceptance

Few respondents noted concerns with acceptance of testing or acceptability of test results, but did reflect serious implications in the potential rollout of the test, especially regarding informed consent and other ethical considerations when weighed against the inherent risk of a pandemic situation. Representative comments included '*The public refusing to take it*' [IC42] and '*comfort for patients*' [IC64] highlighting concerns over the public reaction to testing.

## Time implications

While frequently noted alongside concerns surrounding 'misuse of service' and 'consequences to care', comments directly regarding delays to conveyance as a result of incorporating a SARS-CoV-2 test or other knock-on delays to the service were significant. Some responses followed a patient-centric reasoning —'*ensuring it does not delay transfer, especially in time critical patients*' [IC14], while other comments were concerned about the implications such a delay would have on the ambulance service as a whole—'*delay will increase the time the patient will be with the driver and thus may affect the Key performance Indicator (KPI)*' [IC44] and '*increased job cycle times*' [IC162]. TTR was a key concern among paramedics. This was reflected in the open-response answers *as 'delaying treatment or transport due to waiting for results*' [IC122], '*delays on scene*' [IC113] and '*delaying transport while awaiting results*' [IC83]. In addition, the use of PPE when working with COVID-19 positive patients was noted as having time implications '*Some people wear level 3 RPE for any COVID-19+ve patient, meaning if they tested+ve they would then delay things further by going to change PPE*' [IC161] implying additional time implications beyond the process of administering the test itself and awaiting results although current guidance considers the use of PPE mandatory in any confirmed or suspected COVID-19 case.[21]

## Category 3: practical use of the test

### Test feasibility, correct use and integration with ambulance setting

Non-accuracy and TTR related test characteristics and associated properties such as usability were cited, along with additional direct implications regarding staff training. Size, storage and physical characteristics were commonly referenced in responses—'*How it will be carried—it needs to be introduced into standard response bags that are taken to every patient, similar to our blood sugar testing kits. No one will want to go back to the ambulance to collect a large, bulky box to do it*' [IC34]. Further elaboration regarding the limitations posed by the settings within an ambulance were raised—'*kits won't be stored correctly due to fluctuating temperatures of a vehicle outside in all weathers*' [IC36] and '*keeping in a sterile place*' [IC40]. Issues regarding physical constraints of an ambulance were also invoked though responses '*Quantity & Supply in order to test volume of patients in a shift (~5–9 per 12 hour shift). Availability of tests for relatives*' [IC3] and the ability of users to actually perform the test in such conditions—'*a naso swab could be difficult in a moving vehicle, test would need to be carried out prior to transport*' [IC54].

### Integration into pathways

Test characteristics such as optimum test accuracy and TTRs in order to inform decision making were quoted in a larger number of responses, frequently alongside comments on 'consequences to care' and 'use case to inform clinical decisions'. Aspects of test accuracy were mentioned, either as requirements for a test to be worthwhile—'*rely on the test having very high specificity and less important but high sensitivity*' [IC13], or as a concern of poor accuracy leading to inappropriate clinical decision making—'*That it is not accurate enough but hospitals will begin to rely on it and not follow up test*' [IC101], '*If it is not specific enough it could lead to clinicians making inaccurate decisions about patient care*' [IC107].

## Category 4: safety

### Risk exposure for staff members

Fear of additional harm as a result of testing included not only breaches in personal protective equipment (PPE) as a result of actually conducting the test '*Increased risk to staff carrying out [the] test*' [IC55], '*chance of using AGP's then we could use a L3 mask instead*' [IC129] and additionally the increased time required during call outs while spent conducting the test '*That we will increase our on scene time with a positive patient*' [IC36].

### Staff reassurance

The issue of patient and staff reassurance was covered in a small number of responses—'*would assist in reassurance for crews*' [IC30].

## Follow-up survey 2: optimal test characteristics for prioritised use cases

### Optimal test characteristics for identified clinical scenarios

Respondents were positive about the most accurate hypothetical test (greater than 95% sensitivity and specificity) for use as a decision aid for referral or for triaging of

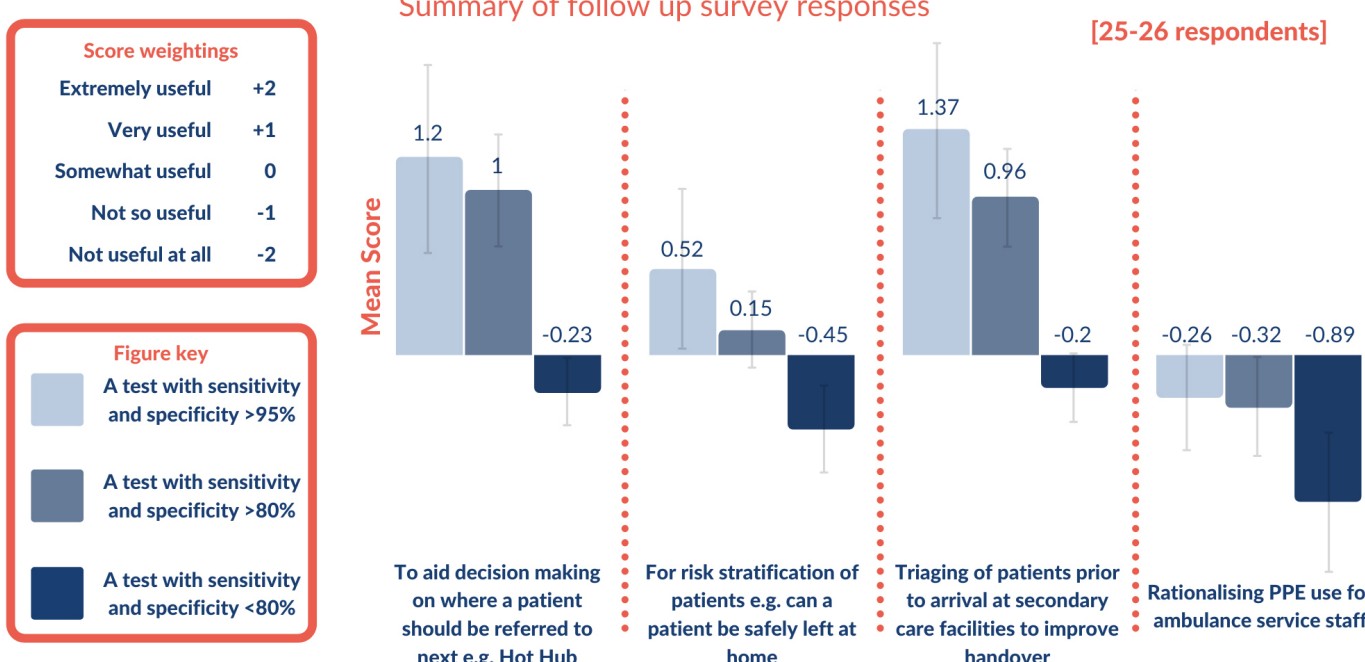

**Figure 3** Summary scores for perceived usefulness of hypothetical SARS-CoV-2 tests for top use cases identified though survey. Summary scores for the Likert ranking of use cases and accuracy statistics for a rapid SARS-CoV-2 test in the ambulance service are displayed in this chart. Likert scores ranged from –2 (Not important) to +2 (Most important). The length of the bars indicates the mean Likert score, while the colour represents the three hypothetical tests referred to in the figure key.

patients prior to arrival at secondary care facilities, but were critical of any test regardless of accuracy being utilised to inform the use of PPE for ambulance staff (figure 3).

Triaging of patients prior to arrival at a secondary care facility was the use case with the most positive responses based on the hypothetical tests presented, with both tests 'over 95% accurate' and tests 'over 80% accurate' receiving an average rating of 'very useful'.

The free-text portion of the follow-up survey was frequently used by respondents to support their answers. The main comments included issues with current ambulance staff compliance with PPE requirements and worry that a negative test may lead to lower compliance with PPE requirements. Some noted that rather than relying on a COVID-19 test, ambulance staff use other clinical indicators to influence decision making, particularly when determining if a patient should be conveyed or kept at home based on standard indicators of illness severity such as pulse and oxygen saturations.

## DISCUSSION

Our results suggest that PoC testing for SARS-CoV-2 within ambulances could add value in particular clinical scenarios. However only an accurate, rapid test would provide the information required to inform clinical decision making. Confidence in the feasibility of rolling out such a test appears limited, an issue which has also been highlighted in other clinical settings.[11] Finger prick blood tests were preferred over other methods, perhaps

supported by respondent's prior experience with blood-glucose tests as highlighted in initial scoping questions on prior experience with PoC testing. Small differences in the operation and use of similar PoC tests mean that training and competency requirements would still be necessary and bring with them additional considerations.[22]

Generally, the most popular use cases for testing focused on patient triage, however, the variation in decision making regarding conveyance to hospital is significant between individual ambulance trusts, and test deployment and response is likely to also vary.[23] Compared with protocol 36-based identification of COVID-19 patients presenting to ambulance services, on-board LFDs may offer a greater positive predictive value and influence more appropriate decision making on triage and conveyance.[24 25] Greater evidence generation into situational use of SARS-CoV-2 diagnostics may reveal patient sub-groups wherein a rapid test influencing the decision making and conveyance of the patient may improve care without substantially impacting the workload and effectiveness of the ambulance service, but significant concerns raised by ambulance service staff would need to be addressed to ensure adherence.

Pragmatically, the TTR of LFDs for SARS-CoV-2 are typically in the range of 15–30 min, fitting with the desired characteristics for paramedics as well as national guidance on reducing hospital admission delays.[26] However, the accuracy they achieve in practice is less than the sensitivity and specificity preferred by respondents to facilitate decision making. The gain in accuracy from typical molecular

assays may not have a TTR within the 30 min window desired by the majority of the respondents. However some molecular assays are available with faster TTR which could be evaluated in this setting.[27 28] An acceptable TTR alone may not alleviate the main concerns raised by ambulance service staff around misuse of the service, additional training requirements and capacity. Recent publication of service evaluation results for on-board LFD use by paramedics has offered promising results on the accuracy of some LFDs in an in-motion ambulance setting (sensitivity 78%, specificity 100%, TTR 10 min) when compared with subsequent PCR testing.[25]

Misuse of service leading to increased workload is clearly an important concern, although there is no evidence yet that members of the public have called emergency services to specifically obtain a test. The overwhelming volume of responses with these concerns, from geographically diverse regions suggests that these views are shared across many ambulance trusts around the UK. This may have been a reflection of the pandemic situation during the conducting of the survey, where ambulance services were experiencing particularly high handover times and some Accident and Emergency departments operating at capacity.[29] It is possible that issues and concerns regarding misuse of the service for the sole purpose of obtaining SARS-CoV-2 tests has been mitigated by the subsequent (5 April 2021) UK government's announcement to make rapid testing available to everyone in England twice weekly.[30]

Overall, the results indicate that ambulance staff are prepared to conduct rapid SARS-CoV-2 testing only if they feel it would add value in making early clinical care decisions, indeed some Ambulance Services trialled the use of on-board lateral flow tests for SARS-CoV-2 at the time this survey work was conducted.[25 31] There were multiple concerns from respondents including the impact of additional testing on speed of care delivery and overall service performance as well as the practical constraints of on-board testing and accuracy requirements. Further research is required to characterise and address these concerns before PoC testing for SARS-CoV-2 can be widely adopted into the ambulance service.

## Limitations

Limited piloting of the surveys was performed due to the short timespan available to develop, disseminate, collect and analyse results, while the short interval of data capture (December 2020–11 January 2021) limited the reach and coverage of the survey. Due to the nature of dissemination, there was wide regional disparity in respondents, with the majority of participants located in the East of England and Greater London, where ambulance waiting times and emergency department capacities were stretched during the period of data collection. The collection of data from all members of the ambulance service, while useful at the time of collection, may mean that applicability of some responses to a first-line use cases in ambulances may not be coming from first-hand experience in that setting. The majority of respondents were front line ambulance staff and paramedics and thus were largely representative of the intended population.

Due to the online interface and open availability, the role and positions of respondents and their understanding of test performance descriptions (ie, sensitivity and specificity) could not be verified. Descriptions and summaries of these words and phrases were included in the survey to ensure understanding. It was evident that some Trusts had been trialling SARS-CoV-2 PoC testing, while others had refrained, meaning that prior experience with such tests was very variable among respondents. The dissemination of the survey primarily through ambulance trust R&D teams and direct contact with respondents may mitigate concerns over false data to an extent.

None of the questions in the surveys were compulsory and most answers provided some free text capacity, those with stronger positive or negative opinions on the subject matter may have been more driven to provide their opinions, introducing a degree of selection bias in participants.

The separation of the questions into two surveys may have introduced bias. Survey 2 had a significantly lower response rate than survey 1 and was only disseminated to those respondents from survey 1 who had agreed to being contacted again.

While some respondents noted experience of using SARS-CoV-2 LFDs in an ambulance setting through trials, there were not enough of these respondents to allow for subgroup analysis of those with experience of using SARS-CoV-2 LFDs, and those without.

**Author affiliations**

[1]NIHR Newcastle In Vitro Diagnostics Co-operative, Translational and Clinical Research Institute, Newcastle University, Newcastle Upon Tyne, UK

[2]NIHR London In Vitro Diagnostics Co-operative, Imperial College London, London, UK

[3]NIHR Newcastle In Vitro Diagnostics Co-operative, Newcastle Upon Tyne Hospitals NHS Foundation Trust, Newcastle Upon Tyne, UK

[4]Centre for Genomics and Child Health, Barts and The London NHS Trust, Blizard Institute, London, UK

[5]Academic Unit of Health Economics, University of Leeds, Leeds, UK

[6]Stroke Research Group, Population Health Sciences, Newcastle University, Newcastle upon Tyne, UK

[7]Institute of Neuroscience, Newcastle University, Newcastle upon Tyne, UK

[8]Healthcare Associated Infections Research Group, University of Leeds, Leeds, UK

**Acknowledgements** The authors would like to thank Graham McClelland for early discussion and scoping for this project, as well as all of the participants who contributed their time and expertise in completing the surveys. The authors are grateful for the input and feedback provided by the MIC Insight Panel and other PPIE members. The authors also acknowledge the support and contributions of the NIHR and CONDOR steering group. Additional members of the CONDOR Steering group include: Dr Emily Adams, Professor Richard Body, Dr Julian Braybrook, Professor Peter Buckle, Professor Paul Dark, Dr Kerrie Davis, Dr Eloïse Cook, Professor Adam Gordon, Mrs Anna Halstead, Dr Charlotte Harden, Dr Colette Inkson, Professor Gail Hayward, Ms Naoko Jones, Professor Dan Lasserson, Dr Joseph Lee, Professor Andrew Lewington, Mrs Mary Logan, Dr Massimo Micocci, Dr Brian Nicholson, Professor Rafael Perera-Salazar, Mr Graham Prestwich, Dr D. Ashley Price, Dr Charles Reynard, Dr Beverley Riley, Professor John Simpson, Mrs Valerie Tate, Dr Philip Turner, Professor Mark Wilcox and Dr Melody Zhifang. The views expressed in this manuscript are those of the authors and not necessarily those of the NIHR or the Department of Health and Social Care.

**Contributors** AJA and JEM conceived the study and design. KG is acting as guarantor and corresponding author. KG, AW, TH and MM contributed to the study design. KG developed the surveys. AJA and MM contributed to survey development. MM and JEM piloted the surveys. JEM, KG and AJA disseminated the survey to Trust R&D teams. KG analysed quantitative data. KG, AJA and MM analysed free-text responses and performed thematic analysis of qualitative responses. TH completed checklist of qualitative reporting. KG and AJA produced the figures and tables. KG and AJA wrote initial manuscript drafts. BS, LS, CP and KD revised and edited the manuscript. All authors reviewed and agreed with the results and conclusions of this manuscript.

**Funding** This study was part of the CONDOR platform (condor-platform.org), which is funded by the UKRI, Asthma UK and the British Lung Foundation (COV0051). KG, AW, TH, CP and AJA are supported by the National Institute for Health Research (NIHR) Newcastle In Vitro Diagnostics Co-operative (http://www.newcastle.mic.nihr.ac.uk/) (MIC-2016-014). MM is supported by the NIHR London In Vitro Diagnostics Co-operative (https://london.ivd.nihr.ac.uk/) (MIC-2016-008). BS is part-funded as an Associate Director of the NIHR Leeds In Vitro Diagnostics Co-operative (https://www.leedsmic.nihr.ac.uk/) (MIC-2016-015).

**Disclaimer** The funders had no role in study design, data collection and analysis, decision to publish, or preparation of the manuscript.

**Competing interests** None declared.

**Patient and public involvement** Patients and/or the public were involved in the design, or conduct, or reporting, or dissemination plans of this research. Refer to the Methods section for further details.

**Patient consent for publication** Not applicable.

**Ethics approval** This work was registered as a service evaluation in Newcastle upon Tyne Hospitals NHS Foundation Trust (project ID: 10641) and granted Caldicott approval (ID: 7867). This study involves no patients or patient information. Local R&D teams approved the dissemination of the survey. Participants gave informed consent to participate in the study before taking part.

**Provenance and peer review** Not commissioned; externally peer reviewed.

**Data availability statement** Data are available on reasonable request. Anonymised data used in this publication is available from the corresponding author on reasonable request.

**ORCID iDs**
Kile Green http://orcid.org/0000-0002-4777-5458
Timothy Hicks http://orcid.org/0000-0002-0942-696X

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
