## [Reviewer comments · BMJ Open]

ARTICLE DETAILS

TITLE (PROVISIONAL)	The perceived feasibility, facilitators and barriers to incorporating point-of-care testing for SARS-CoV-2 into Emergency Medical Services by Ambulance Service staff: A survey-based approach
AUTHORS	Green, Kile; Micocci, Massimo; Hicks, Timothy; Winter, Amanda; Martin, Joanne E; Shinkins, Bethany; Shaw, Lisa; Price, Christopher; Davies, Kerrie; Allen, A

VERSION 1 – REVIEW

REVIEWER	Masuda, Yoshio National University of Singapore, Medicine
REVIEW RETURNED	12-Jun-2022

GENERAL COMMENTS	Thank you for the opportunity to review this manuscript. The paper is an interesting read. This is a topic of interest with much importance especially as it has the potential to improve responses to the pandemic in the context of a frontliner. This paper is also well-written, I only have a few comments as found below: 1. Why is there a need to launch a second survey? I understand that the survey aimed to identify the most acceptable accuracy requirements of rapid tests specifically to meet the prioritised clinical scenarios identified in the first survey. However, I find it more efficient to combine the second survey questions and address it to all listed clinical scenarios from the first survey. Through this method, the response rate would be higher as well. This is further supported by the 52% response rate achieved from the second survey.2. As mentioned in Point 1, the 52% response rate for survey 2 should be mentioned as a limitation in the manuscript, as it introduces bias. Furthermore, only 50 patients out of the original 226 respondents were interested in completing the second survey. This introduces bias as well.3. Could the authors explain why the duration that survey 1 was kept open to responses was different from survey 2?4. I believe the readability of the discussion is good, but most of the discussion focuses around stating the results and is brief on explanation and elaboration, with little references to existing literature. In particular, the paragraph on patient triage (lines 375 to 382), is mostly a call for more research into situational use of SARS-COV-2 diagnostics while bearing in mind the concern for adherence. It would help if the authors expanded on the potential of testing focused on patient triage, the possible benefits behind it, and citing any current experiences from institutions around the world.5. Methodology and reporting standards are valid. Supplementary materials are uploaded and able to be viewed.
---

	6. Line 399: change "It's" to "It is" 7. Line 408: remove "further" 8. Line 409: change "could" to "can"
--	--

REVIEWER	Smit, Liezl Stellenbosch University, Paediatrics and Child Health
REVIEW RETURNED	13-Jul-2022

GENERAL COMMENTS	Thank you for submitting this interesting study that focuses on the perceived feasibility and clinical utility of PoC SARS-CoV-2 testing by paramedics in the UK. I have read the other reviews and author responses before this review and have thus only added additional comments without repeating matters that have already been addressed. Abstract:  • Perceptions instead of opinions? • It is not clear from the abstract if this survey represents the whole of the UK or specific Health Trusts. If a UK-wide study, conclusions based on a small study sample size (from a quantitative perspective) and may not be representative of the population. • No Methods section in Abstract? The abstract gives no indication that this is a mixed -methods survey research project. It is neither clear why 2 surveys were used other than 'follow up' and if the same participants were sampled twice or if completely different participants completed second survey. • Why almost no statistics as part of the results section? Background:  • Well written with clear argument on the knowledge gap addressed by the study. • Write out all abbreviations when first used (e.g. CE-IVD) and use consistent abbreviations (POC vs PoC). Methods:  • Write out all abbreviations (e.g. PPIE, NIHR). • The patient and public involvement section does not make sense as the first information given regarding the methods section. Also not clear if this section is an acknowledgement or a means to address the validity of the survey. What was done to ensure the validity (construct, content, predictive etc) of the surveys? • Survey 1 also included likert scale questions, please add. Results:  • Would be useful for the reader to include more detail on participants apart from geographical area. How many frontline workers, managers (job role) etc? Were there differences in results between these groups? • It is stated in the background section that some trusts have already introduced PoC testing. Were some participants in this study already doing it? Would be interesting to know if those already doing it have a different opinion/perception and whether their concerns are based on actual experiences or hypothetical concerns. • Figure 1. Participants in abstract divided in frontline, call-handlers and management, in figure divided as paramedics and experienced paramedics, technicians and assistance and other? Be consistent.
---

VERSION 1 – AUTHOR RESPONSE

Reviewer 1 Comments	Author Response	Page (P), Line (L) numbers
1. Why is there a need to launch a second survey? I understand that the survey aimed to identify the most acceptable accuracy requirements of rapid tests specifically to meet the prioritised clinical scenarios identified in the first survey. However, I find it more efficient to combine the second survey questions and address it to all listed clinical scenarios from the first survey. Through this method, the response rate would be higher as well. This is further supported by the 52% response rate achieved from the second survey.	Thank you for this. We agree, in retrospect, integrating the questions into a single survey may have resulted in more responses to the Survey 2 questions. At the time of conducting this work ambulance services in the UK were overwhelmed. We wanted to obtain useful and actionable information, whilst keeping the burden on frontline workers to a minimum. We did not expect such a strong response, and a number of respondents to Survey 1 stated that they would be happy to be contacted again for follow-up questions. This prompted us to issue Survey 2 soon afterwards to those who had consented to being re-contacted. We are grateful to the respondents for their participation at such a critical time.	P3, L63-64 P15, L453-455
2. As mentioned in Point 1, the 52% response rate for survey 2 should be mentioned as a limitation in the manuscript, as it introduces bias. Furthermore, only 50 patients out of the original 226 respondents were interested in completing the second survey. This introduces bias as well.	Agreed. This has been added to the limitations and to the new section 'Strengths and Limitations of this study' suggested by the editor.	P3, L63-64 P15, L453-455
3. Could the authors explain why the duration that survey 1 was kept open to responses was different from survey 2?	Survey 1 was kept open for 40 days, with the follow up survey open for 15 days. The initial survey was kept open for approximately 6 weeks so that ambulance service staff could respond in their own time whilst balancing the need for actionable data during the peak of the pandemic. The second survey was open for a much shorter period of time as it was sent only to participants that had agreed to be re-contacted and we wanted to ensure a quick turnaround time.	N/A
4. I believe the readability of the discussion is good, but most of the discussion focuses around stating the results and is brief on explanation and elaboration, with little	Thank you for the comment on readability. We agree with the assessment of the discussion and have added more inference, as suggested. Given the time between conducting the survey and this round of reviewing, there is more published	P13-P14, L378-428

references to existing literature. In particular, the paragraph on patient triage (lines 375 to 382), is mostly a call for more research into situational use of SARS-COV-2 diagnostics while bearing in mind the concern for adherence. It would help if the authors expanded on the potential of testing focused on patient triage, the possible benefits behind it, and citing any current experiences from institutions around the world.	literature available which has now been referenced. Please see discussion and tracked edits.	
5. Methodology and reporting standards are valid. Supplementary materials are uploaded and able to be viewed.	Thank you for this comment. We appreciate the time you have taken to review this manuscript.	N/A
6. Line 399: change "It's" to "It is"	Thank you for highlighting this error. It has been corrected in the revised manuscript	Table 2, P10,L262
7. Line 408: remove "further"	Thank you for highlighting this error. It has been corrected in the revised manuscript	P13,L427-428
8. Line 409: change "could" to "can"	Thank you for highlighting this error. It has been corrected in the revised manuscript	P13,L428
Reviewer 2 Comments	Author Response	Page (P), Line (L) numbers
Abstract: 1 Perceptions instead of opinions? 2 It is not clear from the abstract if this survey represents the whole of the UK or specific Health Trusts. If a UK-wide study, conclusions based on a small study sample size (from a quantitative perspective) and may not be representative of the population. 3 No Methods section in Abstract? The abstract gives no indication that this is a mixed -methods survey research project. It is neither clear why 2 surveys were used other than	 1. Agreed. Edited in revised manuscript. 2. The Survey was disseminated via individual Trust's R&D services at their discretion, which resulted in considerable variance in response rates across the regions. The survey results are likely not representative of the population, but do offer some insights into frontline worker's concerns. This has been noted in the limitations section of the manuscript. 3. Sorry for this oversight. 'Methods' is not one of the abstract sections in the submission guidelines for this journal. We have noted the methods under the 'Design' sub-heading in the revised abstract, but are limited by word count. 4. We did not want to assign quantitative analysis to the qualitative components of the survey as this was not part of the initial project plan. Summary statistics (numbers 	1.P1, L22 2.P1,L27-28; P15, L430-440 3.P1, L25-26 4.P1-2, L31-42

'follow up' and if the same participants were sampled twice or if completely different participants completed second survey. 4 Why almost no statistics as part of the results section?	and percentages) have been included where appropriate.	
Background: 1 Well written with clear argument on the knowledge gap addressed by the study. 2 Write out all abbreviations when first used (e.g. CE-IVD) and use consistent abbreviations (POC vs PoC).	 We thank the reviewer for these comments and appreciate the time taken to review this manuscript. Thank you for highlighting these errors. We have edited and corrected for consistency in the revised manuscript 	1.N/A 2.Throughout. Please see uploaded version with tracked changes.
Methods: 1 Write out all abbreviations (e.g. PPIE, NIHR). 2 The patient and public involvement section does not make sense as the first information given regarding the methods section. Also not clear if this section is an acknowledgement or a means to address the validity of the survey. What was done to ensure the validity (construct, content, predictive etc) of the surveys? 3 Survey 1 also included likert scale questions, please add.	 Thank you for highlighting this error. It has been corrected in the revised manuscript Thank you for this comment. We included PPIE as the first section of the methods to highlight the importance of PPIE in our work and in recognition of the time and effort provided by our PPIE team. The content of this section was based on the submission guidelines for this journal [https://bmjopen.bmj.com/pages/authors/#submission_guidelines]. The language and content of the surveys was assessed through the patient and public panel. Additionally, piloting of the survey was performed by the research group and external clinical experts to ensure it would work as intended. Results gathered during the piloting of the survey were not included in the manuscript. We have restructured this section to address the reviewer comments and moved it to the end of the methods section. Apologies, this information is under the 'Purpose & Structure' and 'Data Analysis' sub-headings of the methods section. 	1.Throughout & in 'list of abbreviations' P15-16, L460-476 2.P6, L184-194 3.P5, L143-160; P6, L167-194
Results: 1 Would be useful for the reader to include more detail on participants apart from geographical area. How many frontline workers, managers (job role) etc? Were there differences in results between these groups? 2 It is stated in the background section that some trusts have already introduced PoC testing. Were some participants in	 We have added an expanded table of demographics to highlight job roles of participants by region [Table 1]. Yes. A small number of paramedic respondents noted that they had been involved in short trials utilising LFTs on-board their ambulances. There were not enough of these respondents to allow for sub-group analysis and the trials were still in progress at the time of conducting the survey. We have added this to limitations section. Apologies for this oversight. This has been adjusted for consistency across the tables, figures and content. 	1.Table 1, P7, L206 2.P15,L456-458 3. Throughout. Please see uploaded version with tracked changes.

this study already doing it? Would be interesting to know if those already doing it have a different opinion/perception and whether their concerns are based on actual experiences or hypothetical concerns. 3 Figure 1. Participants in abstract divided in frontline, call-handlers and management, in figure divided as paramedics and experienced paramedics, technicians and assistance and other? Be consistent.		
--	--	--

VERSION 2 – REVIEW

REVIEWER	Masuda, Yoshio National University of Singapore, Medicine
REVIEW RETURNED	19-Sep-2022

GENERAL COMMENTS	Thank you for addressing my concerns. I believe that the corrections have adequately improved the manuscript. However, my concern regarding the integration of the questions into a single survey remains. This introduces bias and also affects the respondents rate on results that the study is based on. On one hand, it may be viewed as an error in the study design, but on the other hand, this can be expected in survey-based studies conducted in time critical conditions such as during the pandemic. Moving forward, if the authors decide to conduct follow-up studies, the authors should take note of this and modify their study design.
--

REVIEWER	Smit, Liezl Stellenbosch University, Paediatrics and Child Health
REVIEW RETURNED	14-Sep-2022

GENERAL COMMENTS	This is an important study that highlights the need to pay careful attention to the human factor for successful implementation of a quality improvement initiative.
---